# Improving Quality in the Process of Hot Rolling of Steel Sheets

**Stefan Markulik \***, **Anna Nagyova** , **Renata Turisova** and **Tomas Villinsky**

Department of Safety and Quality, Faculty of Mechanical Engineering, Technical University of Kosice, Letna 1/9, 04200 Kosice-Sever, Slovakia; anna.nagyova@tuke.sk (A.N.); renata.turisova@tuke.sk (R.T.); tomas.vilinsky@student.tuke.sk (T.V.)

\* Correspondence: stefan.markulik@tuke.sk; Tel.: +421-948-881-988

**Abstract:** The hot rolling of steel sheets is a highly energy-intensive process. There are technical and operational issues associated with this process, and the causes of these issues can be various. This study involved analysis of one issue that has a great influence on the resulting surface quality of rolled sheet metal: rolled foreign material. After the sheet cools, rolled foreign material tends to fall off and a hole then remain on the surface of the sheet. This paper focuses on the search for the root causes of the occurrence of foreign material rolling. The basic categorization of the causes of this issue was performed by experienced long-term operators. The 4M method (man, machine, method, and material) was used to categorize the causes. Pairwise comparison was used to verify the result. Using energy dispersive spectroscopy analysis, the origin of the foreign material was identified. The analysis confirmed that the foreign material was not derived from the primary material. Further research showed that the cause of the issue was the guide rulers, which are a structural part of the rolling mill. Measures were taken to significantly reduce the incidence of the problem, which also had the effect of reducing financial losses, which fell by a third in 18 weeks. In the future, it will be necessary to make design changes (modernization of the rolling mill), which will, however, require more financial investment.

**Keywords:** hot rolling; analysis; cause; rolled foreign material

## 1. Introduction

The automotive industry is currently one of the largest processors of rolled sheets. It is an industry that is ready to integrate the principles of Industry 4.0 into its structures [1]. In the industrial production sector, the demand for a wider range of products in ever-increasing quantities is constantly growing. The production of steel must be based on predetermined properties required by the customer to meet demanding specifications [2].

In the professional literature, increased attention has been paid to the analysis of the relationships between human and machine [3] by manipulating the device, manipulating the material, and understanding the causes of the individual errors [4]. Ikumapayi at al. [5] focused on an approach for rolling techniques in a metal forming operation as part of an industrial production process. Developmental trends in metal forming, as well as the existing progress achieved in metal forming processes, were presented in [6]. The proposal of hot rolling as a method of metal forming was published in [7]. The aim of this paper is to increase the quality of the process of hot rolling of steel sheets with a focus on reducing various defects, which will subsequently be reflected in reduced product quality, or in the form of increased costs associated with the process.

Highly efficient and fully automated lines need to have certain conditions met for their operation. Adhering to them in real practice is not easy. For example, although the sheet may appear perfectly flat, the distribution of the residual stress may be so disadvantageous that it leads to flatness defects. The material properties of the metal strip can also be changed along its length, which corresponds to the need for a suitable adjustment of the equalizer.

Grüber & Hirt [8] proposed a numerical model of a seven-cylinder equalizer to link the target values of plate flatness and residual stress distribution to determine the positions of the cylinders, resulting in a flat plate and defined residual stress distribution. Using numerical calculations, they investigated the general potential to change the distribution of the residual stress by levelling the cylinder. Surface defects in hot rolling were examined in [9]. The authors proposed a procedure by which it was possible to increase the accuracy of the classification of the mentioned defects using a support vector classification scheme based on process knowledge. Even with small deformations in the hot rolling process, metals and alloys are subjected to complex stress, strain, strain rate, stress flow, tensile stress, tensile pressure, and pressure. The reduction in said deformations usually requires complex analytical methods. The study of defects arising from small deformations by hot rolling is included in [10]. The output of the hot rolling process should be a quality rolled product with the required uniform accuracy, a defect-less surface, and the expected mechanical properties. The correct setting of individual process parameters is a crucial requirement for meeting the expected high quality, as well as geometric properties, of rolled products. The issue of hot rolling is in itself very complex. It is a process that depends significantly on the flow of metal at high temperatures, on strong plastic deformations during molding, and on other physicochemical processes. The large number of variables that need to be considered, as well as the difficulties associated with measuring the individual characteristics of the process, significantly affect this complexity. However, experience has shown that the operator has a significant impact on the quality of the rolling process, even though its individual processes are largely standardized. Bordonaro at al. [11] used finite element methods and an experimental planning method to examine the flat rolling process according to a series including parameters and scenarios. Using the Pareto analysis described in [12], the authors investigated the optimal combination of factors to ensure the maximum fulfillment of goals.

In this paper, we focus on specific errors in the hot rolling process. The subject of our research is the gluing of foreign material on the guide rules to the individual stools of the order of finish.

## 2. Materials and Methods

Hot rolled coils are among the basic raw materials used in the manufacturing industry. These coils are used to produce pipes, floors, furniture, ships, buildings, and agricultural machinery. In recent decades, there has been a strong expansion in the production of car body parts with specific strength requirements [13]. Heating furnaces are used in the production of coils in a hot rolling mill. They are also used to heat steel semi-finished products (sheet bars, slabs, and ingots) to a temperature of about 1200 °C. This temperature is suitable for the plastic deformation of steel. Hot rolling is a process in which the plastic deformation of said steel semi-finished products occurs. The heating process in the heating furnace is a continuous process. At the entrance of the furnace, blooms with a temperature between 100 and 250 °C are inserted. Heat is transferred to the steel material during the passage through the furnace mainly by convection and radiation from gas burners from the furnace walls [14].

The main production unit is the rolling mill. This represents the grouping of different numbers of rolling stands into a production line. The number of rolling stands can vary from one to twelve and can be higher if necessary. Lines for coarse billets usually have a smaller number of rolling stands and larger removals, and the operations are performed by repeating the course while constantly adjusting the rolling gap. Lines for fine billets and rolling have a larger number of rolling stands. Rolling mill contains various equipment (e.g., material movement, heating, cutting, straightening, etc.) [15].

Defects are classified according to different criteria. For the purposes of this work, they were classified into the following categories: technological, material, and other defects. Technological errors include defects caused by the design of equipment where the surface of the strip is mechanically damaged, as well as errors caused by the technological process

of production itself, such as low temperature, poor speed, and poorly set rolling forces [16]. Material defects are understood to signify defects of metallurgical origin, i.e., the influence of the chemical material composition. They can be caused by improper handling or defects during the transport of material [17]. Other errors include the effects of ambient temperature, energy supply effects, line downtime, humidity, aggressive environment, or human factors [18]. In the case of rolled foreign material, it is difficult to determine which category it clearly belongs to. This is a technological and material defect. The cause results from the nature of the technological process (technology of rolling). Rolled foreign material can be referred to as a material (element) that has entered the primary material during the rolling process. It is characterized by a different chemical composition or other mechanical properties, and its origin is different from the origin of the primary rolled material. This is a material that is not related to the primary material. Such foreign materials may be, for example, scale, foreign objects (such as screws and nuts), or a fragment from a loose part of the rolling mill (e.g., pieces of pertinax plates).

### 3. Error Analysis of Hot Rolling

In this study, the primary task was to identify the location of errors in the rolling process. First, the visual manifestation of the error was examined. On this basis, it was possible to determine whether the error occurred during the cold rolling process or the hot rolling process. In Figure 1, a photograph of a strip of sheet metal after cold rolling is shown. The red box indicates the location of the rolled foreign material. Due to the visual manifestation of the defect (foreign rolled material is stretched), the suggestion that it could be rolled during the cold rolling process can be excluded. Further analysis focused on the hot rolling process.

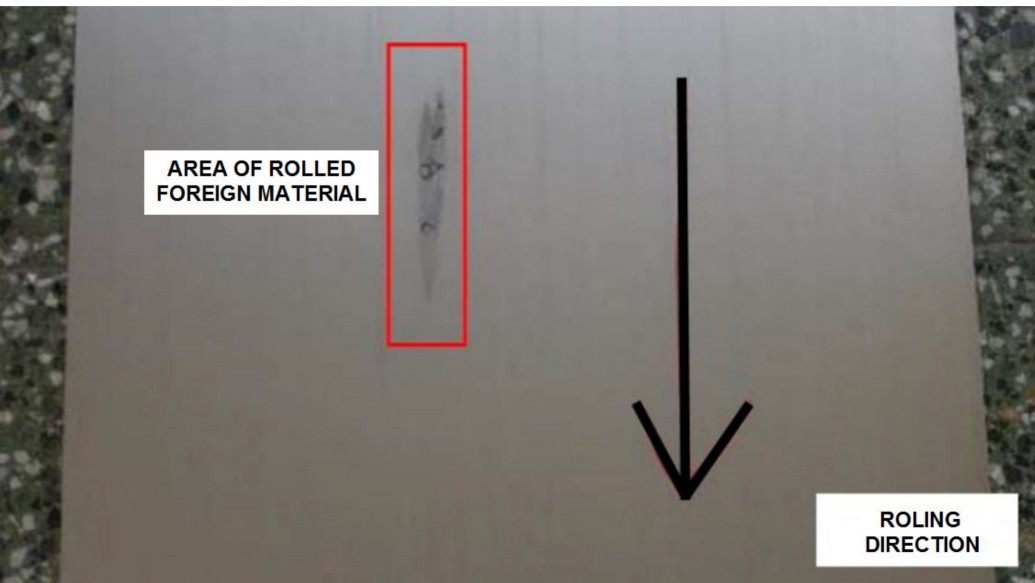

**Figure 1.** Occurrence of a strip error. Source: own research.

A suitable analysis for determining root causes is the cause-and-effect analysis [19,20]. The assessment of the individual causes should be based on the experience of workers [21]. The root causes of the problem were identified, and the analysis was performed by five employees (operators). Each of them was demanded to assign points from 1 to 9. The highest point rating (9 points) was assigned to the potential cause that was most probable from his point of view. One point was assigned to the least likely one. The processed evaluation results (scores) and categorization of causes into 4M (man, machine, material, method) groups are shown in Table 1.

**Table 1.** Processing of potential causes—4M categories. Source: own research.

| Category (4M) | Factor | Cause | Evaluation from Operators | | | | | SCORE |
|---|---|---|---|---|---|---|---|---|
| | | | $O_1$ | $O_2$ | $O_3$ | $O_4$ | $O_5$ | |
| **Man** | Technologist | Lack of knowledge | 1 | 3 | 2 | 1 | 1 | **8** |
| | | Failure to follow the procedure | 4 | 4 | 4 | 8 | 4 | **24** |
| **Machine** | Rolling mill | Old rolling mill construction | 9 | 7 | 7 | 6 | 8 | **37** |
| | Guiding rules | Unsuitable guide rail material | 8 | 9 | 8 | 9 | 9 | **43** |
| **Material** | Rolled material | Chemical purity | 6 | 6 | 9 | 5 | 6 | **32** |
| **Method** | Rolling speed | Undefined specification | 2 | 1 | 1 | 2 | 2 | **8** |
| | Rolling temperature | Absence of temperature measurement | 3 | 2 | 3 | 3 | 3 | **14** |
| | Rolling gap | Undefined specification | 5 | 5 | 5 | 4 | 5 | **24** |
| | Cleanliness of the environment | The absence of a directive on cleaning | 7 | 8 | 6 | 7 | 7 | **35** |

The results obtained from this analysis indicate the order of potential causes. The highest score was achieved by the "Unsuitable guide rail material" response (43 points), followed by the "Old rolling mill construction" response (37 points).

To obtain a valid result, a method for determining the weights of the criteria based on pairwise comparison was used. This aimed to determine the preferential relationships of criteria–cause pairs. For each criterion, the number of preferences for all others was determined. Pairwise comparison was used to validate the results [22]. The pairwise comparison method is sometimes referred to as Fuller's triangle [23]. The aim was to determine the weights of the criteria–causes listed in Table 1 by means of pairwise comparison. For each pair of criteria, operators determined whether they preferred the cause listed in the row to the cause listed in the column. If so, the value 1 was written in the appropriate cell of Fuller's triangle. Otherwise, the value 0 was entered (Table 2). For each cause, the value of preferences was determined. This is equal to the sum of those entered in the line of the given cause and the sum of zeros in the column of this cause. The normalized weight of the criterion–cause was calculated according to the relation [23]:

$$v_i = \frac{f_i}{\sum_{i=1}^{n} f_i} \tag{1}$$

Whereas the feasible comparison is given by the following relationship:

$$\sum_{i=1}^{n} f_i = \frac{n.(n-1)}{2} \tag{2}$$

where

$v_i$—Standard weight of the *i*-th criterion (causes);
$f_i$—Number of preferences;
$n$—Number of criteria (causes).

The determination of preferences was carried out by all five operators together. Operators always agreed on one joint assessment. Pairwise evaluation, determination of preferences, and weights of individual causes are given in Table 2 whereas the feasible comparison is given by the following relationship in equation (2).

It can be stated that the results obtained using the score (Table 1) and the calculated standard weight (Table 2) match. It follows that the results given in Table 1 are valid [21], i.e., they can be used for further analysis [23].

If a given strip of sheet is rolled to a thickness of about 0.2 mm, the foreign material falls out and a hole is formed over the entire thickness of the sheet. In these places, the cross-section of the strip is weakened locally by rolling inhomogeneous (foreign) materials, due to which stresses may exceed the strength of the material, causing it to break.

An example of foreign material can be seen in Figure 2. It is not connected to the surrounding material, and at the same time, it is completely rolled straight from the top. No pushing out occurred on the underside of the primary material. This proves that the place of origin is the hot rolling process.

**Table 2.** Pairwise comparison of causes. Source: own research.

|  |  | $C_1$ | $C_2$ | $C_3$ | $C_4$ | $C_5$ | $C_6$ | $C_7$ | $C_8$ | $C_9$ | Preferences | Weight |
|---|---|---|---|---|---|---|---|---|---|---|---|---|
| Lack of knowledge | $C_1$ |  | 1 | 0 | 0 | 0 | 0 | 0 | 0 | 0 | 1 | 0.0278 |
| Failure to follow the procedure | $C_2$ |  |  | 0 | 0 | 1 | 0 | 0 | 1 | 1 | 3 | 0.0833 |
| Old rolling mill construction | $C_3$ |  |  |  | 0 | 1 | 1 | 1 | 1 | 1 | 7 | 0.1944 |
| Unsuitable guide rail material | $C_4$ |  |  |  |  | 1 | 1 | 1 | 1 | 1 | 8 | 0.2222 |
| Chemical purity | $C_5$ |  |  |  |  |  | 1 | 1 | 1 | 0 | 4 | 0.1111 |
| Undefined specification | $C_6$ |  |  |  |  |  |  | 0 | 0 | 0 | 2 | 0.0556 |
| Absence of temperature measurement | $C_7$ |  |  |  |  |  |  |  | 0 | 0 | 3 | 0.0833 |
| Undefined specification | $C_8$ |  |  |  |  |  |  |  |  | 0 | 3 | 0.0833 |
| The absence of a Directive on cleaning | $C_9$ |  |  |  |  |  |  |  |  |  | 5 | 0.1389 |
|  |  | | | | Number of comparisons: | | | | | | 36 | Σ 1 |

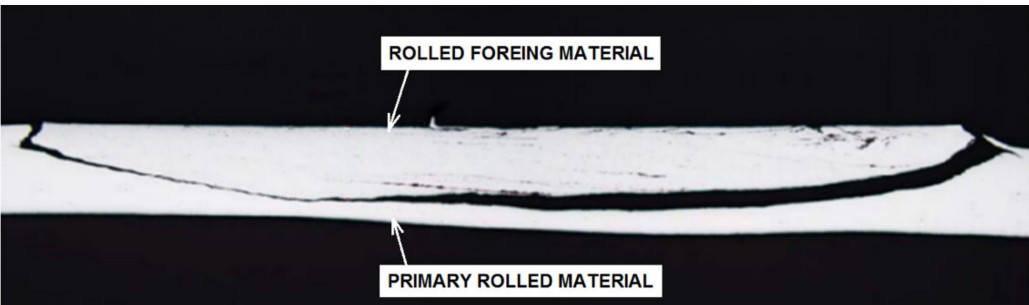

**Figure 2.** Microstructure of rolled foreign material. Source: own research.

The next step in analyzing the origin of the foreign material was to focus on its chemical composition. This made it possible to determine the origin of the rolled material. This means determining whether it comes from the primary material (from the material being rolled) or not.

### 3.1. Energy Dispersive Spectroscopy Analysis

Energy dispersive spectroscopy (EDS) analysis was used to determine the chemical composition of the foreign material. The principle of this method is based on the reflection of X-rays with kinetic energy of keV (kilo-electron volt). In [24], the analysis was also used to reveal microstructure details. In [25], the X-ray Li K in lithium compounds was detected using EDS analysis. The use of EDS analysis in our own research did not confirm the origin of the foreign material derived from the primary material. The chemical composition of the foreign and primary material differs.

In Figure 3 (left side), the arrow indicates the location analyzed. It is clear from the figure that this differs from the surrounding material. The second part of the figure shows the representation of the chemical elements carbon (C), oxygen (O), and iron ($\gamma$-Fe and $\alpha$-Fe). This proves that these are steels from the hot rolling process. The hot rolling process was determined by identifying the location of the problem. In this way, the individual parts of the hot rolling mill line had to be analyzed. The question arose as to whether the occurrence and rolling of foreign material could be caused by rotating or static parts. The influence of rotating parts was excluded due to the occurrence of the rolling of foreign material. If an error were caused by a rotating part, its occurrence on the rolled strip would have to be caught repeatedly [26]. This has not yet been confirmed. By excluding rotating parts, the large number of possible causes of the error were significantly reduced. In the

case of static parts (from blast furnaces to winders), only two were under consideration, namely:

- Guide rail rulers—In preparation order and order of finish.
- Slide bars in blast furnaces.

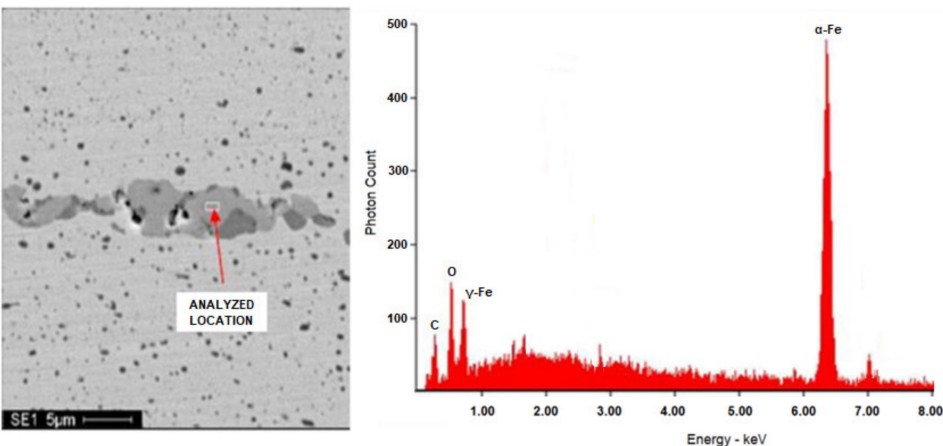

**Figure 3.** EDS analysis. Source: own research.

### 3.2. Analysis of Guide Rulers

The rolling mill is divided into a preparatory order and an order of finish. Both parts contain guide rulers. The rulers are guided symmetrically around the central axis of the rolling mill. They are set to the width of the billet, with added clearance offset. The offset of the rulers was adjusted to make it easier to insert saber-shaped and wide billet beginnings. The offset size was set by the line control system or operator. The position of the rulers was adjusted by means of an electric motor with opposing guide screws and nuts. After visual inspection of the rulers of the preparatory order and order of finish, the following were found:

(a) In the preparatory order, the rolled material did not stick to the guide rulers. This is not due to the low rolling speed and the large thickness of the material.

(b) The rolled material was glued in the order of finish of the hot rolling mill. The strip led to the stools rubbing against the rulers. The cause of this is the high temperature of the material (it is soft) and the contact surface not being perfectly smooth. The rolled material was glued to the rulers (Figure 4).

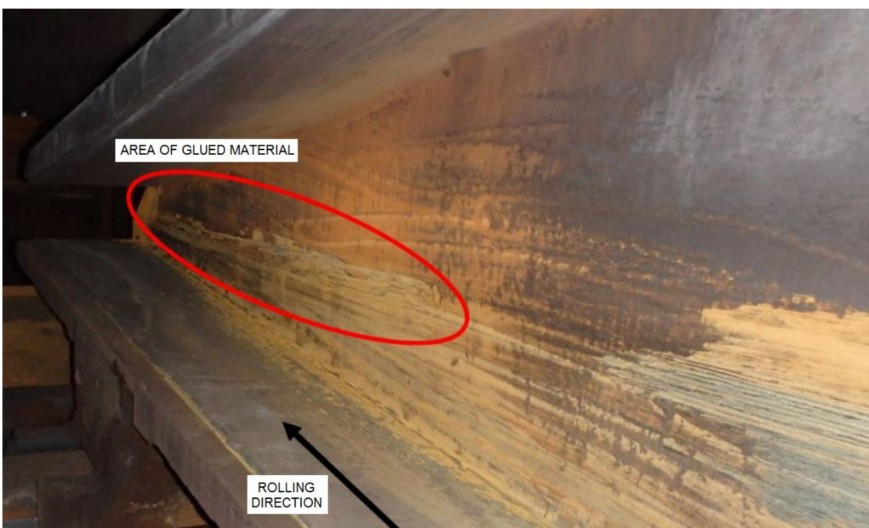

**Figure 4.** Glued material on guide rulers. Source: own research.

To ensure that the rolled strip is guided correctly, the rulers must not have an offset that is too large. Sometimes, a situation where the offset is not large enough can occur. Then, the rolled strip can seize the ruler and create grooves in it. After successfully locating the site related to the formation of foreign material, further steps were identified. This constituted an analysis of different types of factors, such as the work shift (human factor), width of the rolled strip, and thickness of the rolled strip.

### 3.3. Analysis of the Influence of the Human Factor

The analysis of the influence of the human factor included an examination of individual work shifts. The analysis of work shifts (Figure 5) showed that the frequency of errors was influenced by individual work changes. The results show that the lowest occurrence of holes was observed during work shift C. The difference between the lowest and highest incidence between changes was more than 60%.

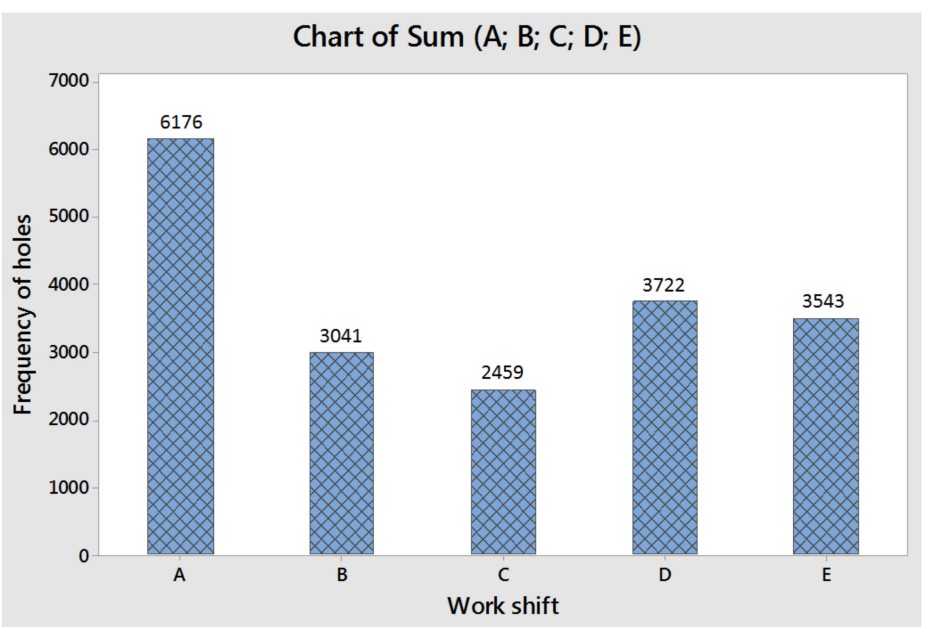

**Figure 5.** Occurrence of holes according to work shifts. Source: own research.

### 3.4. Analysis of molding of the Finish Stands

The aim of this analysis was to determine the clearance (distance) of stands. This means whether the window in the stand is within the tolerance or outside the prescribed specification. The consequences of non-compliance with the tolerance can be manifested by crossing the rollers. The crossing can be perpendicular to the rolling axis or in the same direction as the rolling axis. Crossing of the rollers can affect high convexity, uneven strip thickness, or unsatisfactory wedging. Even if the tip of the strip is in the axis of the rolling mill, the body of the strip may still rub against the guide rulers. This can also be a consequence of the crossing of the rollers and the technical condition of the stand bars of rolling stands of the order of finish. Figure 6 shows the finish stands in the order of production flow. The height of the bars in the graphs shows the percentage of values in accordance with Table 3.

Figure 6 shows that the worst case ($\geq 0.81$ mm—Values in nonconformity) was on rolling stands H6 and H8. Rolling stand H11 also showed poor conditions. Values $\geq 0.81$ mm show a level of more than 50%, and values in the range $0.66 - 0.8$ mm reached a level of 35%.

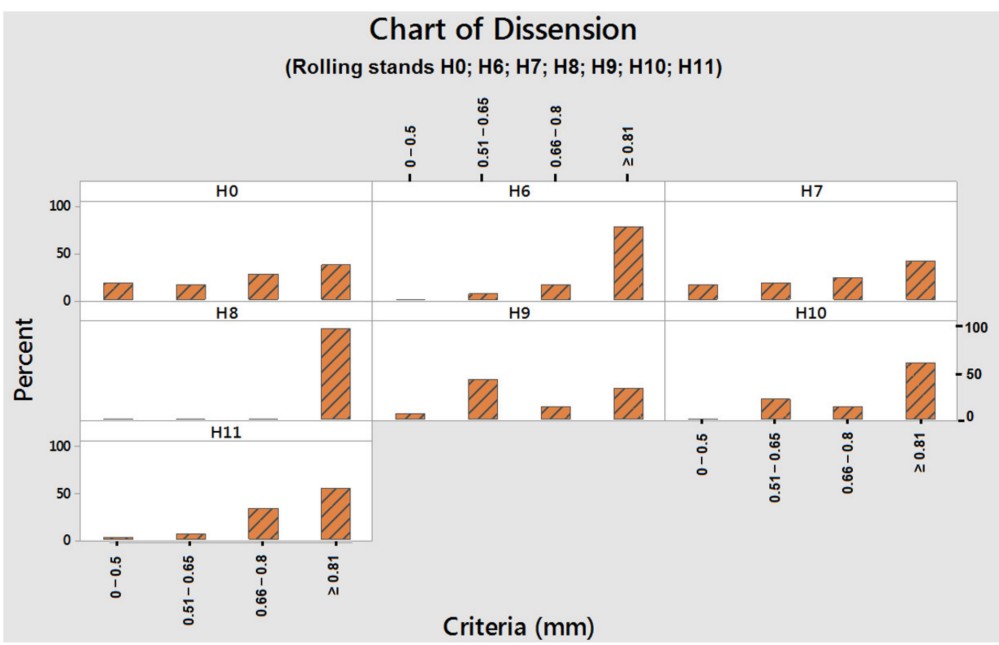

**Figure 6.** Molding of stands—Overall. Source: own research.

**Table 3.** Production specification (for Figure 6). Source: own research.

| 0.0–0.5 mm | Values in full conformity |
|---|---|
| 0.51–0.65 mm | Values in conformity (close to partial conformity condition) |
| 0.66–0.8 mm | Values in partial conformity |
| ≥0.81 mm | Values in nonconformity |

## 4. Discussion

Based on the analyses performed, the following measures were taken:

(a)    Change in the guide rail material

An experimental design based on all the analyses described above aimed to change the material of the guide rails. The rulers were a place where foreign material accumulated. After its accumulation, it loosened and was subsequently rolled during the rolling of the primary material. Changing the material of the guide rails could eliminate this issue. The request to change the material of the rulers was addressed to an external supplier. The basic requirements included higher abrasion resistance at high temperatures, high resistance to cyclic heat loading, higher hardness, and, at the same time, sufficient toughness. Under such requirements, the properties of the rulers would ensure their long life while significantly reducing the occurrence of rolling of foreign material. The requirements for the properties of the rulers were demanding. After 3 months, the external supplier supplied new rulers for experimental testing. Testing took place with increased control of their technical condition. After 3 days of use, cracks appeared on the rulers, which were not technologically acceptable. A detailed image of a crack is shown in Figure 7, and after analysis, it was found that this crack travelled transversely in the direction of rolling over the entire surface of the ruler, and was not even visible in some parts, i.e., it was distributed within the material.

This measure (change in the material of the guide rulers) was perceived from the beginning as an exceptionally good measure. However, the new guide ruler broke after three days of operation of the rolling mill and became unsatisfactory (Figure 7). This measure was experimental because the guide supplier had to supply new rulers made of a different material than the original ones. As this measure was an experiment and the economic possibilities of the organization were limited, the management decided to stop

searching for more material, as it would require a considerable financial investment with an uncertain outcome.

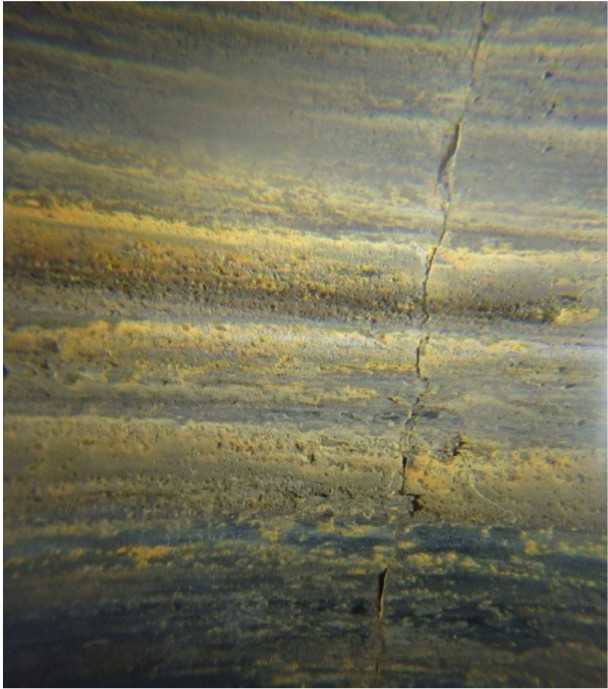

**Figure 7.** Crack of a guide ruler. Source: own research.

(b)    Measures relating to the human factor

(*b1*) Based on the analysis of individual work shifts, it was found that work shift C had the best results in terms of the lowest occurrence of the rolling of foreign material. It was agreed with the internal logistics department that the work shifts would be adjusted in view of the above.

The measure was implemented, but an assessment of its effectiveness has not yet been carried out.

(*b2*) Analysis of the stand bars of the order of finish showed the worst technical conditions on rolling mills H6 and H8 (Figure 6). The measure aimed to replace all of the bars on these two stands. Routine maintenance activities were performed on the other stands with respect to time and financial limits.

The measure was introduced and induced a significant improvement in the technical condition of the rolling mill. The holes were reduced due to the rolling of foreign material. At the same time, other errors were reduced. Evidence of technical improvement is shown in Figure 8 (compared to Figure 6). The effectiveness assessment was carried out 18 weeks after the introduction of this measure.

When assessing the occurrence of errors after the proposed measures (Figure 9), a significant decrease in the incidence of errors (i.e., holes due to rolled foreign material) was noticed. The decrease represents a value of up to 33%.

The measures taken reduced the material loss of steel sheet production. The loss before and after taking the measures is shown in Table 4.

**Table 4.** Effect of the measures. Source: own research.

| Loss before Measures (Tons) | Loss after Measures (Tons) | Reduction (%) |
| --- | --- | --- |
| 1668.36 | 1115.09 | 33.16 |

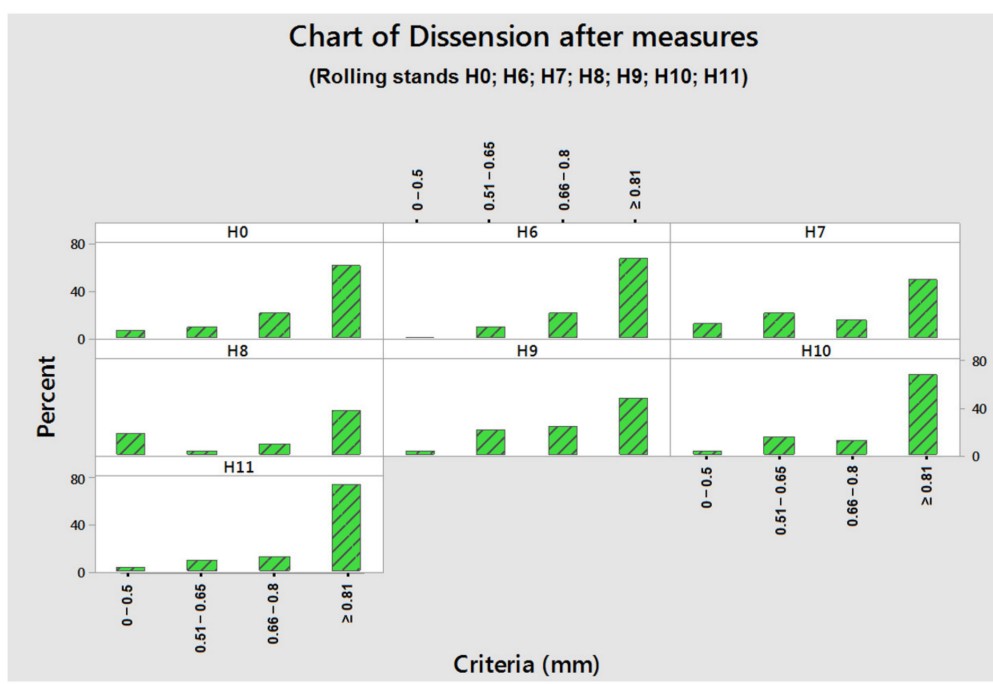

**Figure 8.** Bars of stands after measures—Overall. Source: own research.

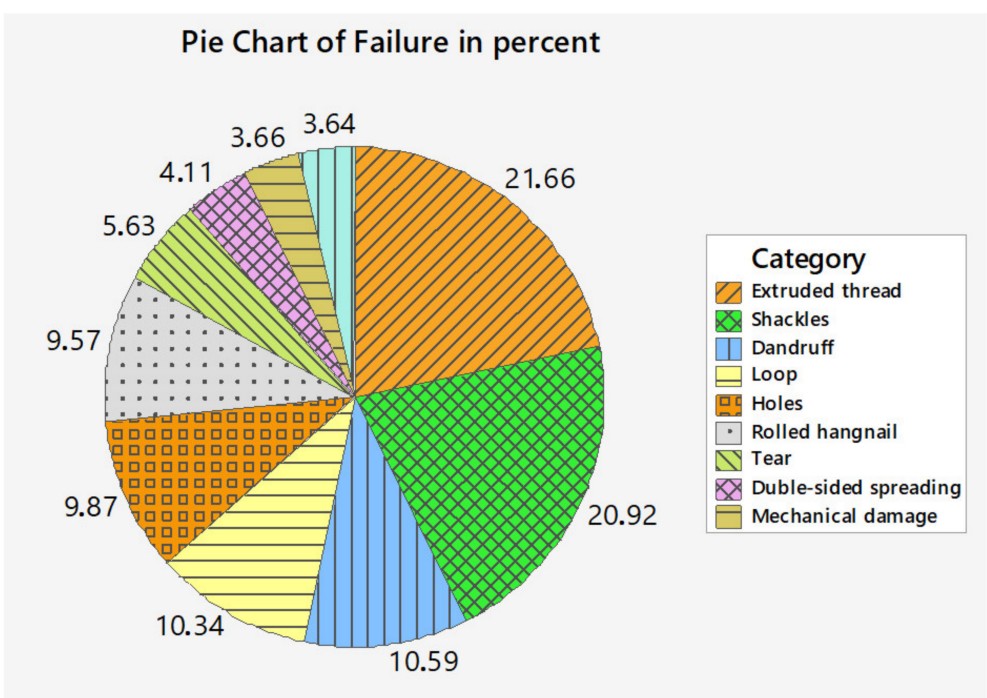

**Figure 9.** Occurrence of errors after measures. Source: own research.

## 5. Conclusions

The aim of this research was to identify the main causes of the occurrence of rolled-in foreign material. Finding the root causes of any problem requires the use of several tools and analyses. The need for such an approach is important to verify the results obtained from the previous steps of the research. The research has shown that the use of quality tools helps to identify the potential causes of the problem under analysis [27,28]. When solving a problem, it is important to find out the real cause of the problem. An incorrectly identified cause may in fact only be a manifestation of the root cause [29]. By analyzing the results

obtained by individual analyses, the root cause can be determined [30–32]. In this research, based on the activities described, it was possible to propose measures to improve the hot rolling process. Knowledge of the organization and analytical methods were used [33]. Statistical analyses of the factors that may have caused the occurrence of rolled-in foreign material were performed to achieve the stated objective. The key step was to correctly and accurately identify the location of the defect stated in this research. Subsequently, it was possible to propose appropriate measures based on the experience and skills of the workers. Corrective actions were taken. Some were rejected (e.g., repeated replacement of a new guide rulers or finding a more durable material for a new guide rulers) as they required significant costs. This led to an analysis of the stand rails of the finishing line. The research showed that their technical condition has a direct and significant influence on the occurrence of the observed rolled-in foreign material. Thanks to properly designed corrective measures, the technical condition of the hot rolling mill's finishing line has been improved. The total material saving on production, expressed in financial terms, was EUR 305,000.00 in 18 weeks. The knowledge gained from the research may be useful for other organizations facing similar problems and a also having limited financial resources for a comprehensive refurbishment of the hot rolling mill. In general, the research findings will help to demonstrate how important it is to determine the root cause before taking measures to manage the problem under analysis.

**Author Contributions:** Management and validation, writing and final review, S.M. and T.V.; application of statistical, mathematical techniques, R.T.; research and verification, S.M.; development of methodology, S.M. and T.V.; data collection, A.N. and T.V. All authors have read and agreed to the published version of the manuscript.

**Funding:** This contribution is the result of the implementation of the following projects: KEGA No. 019TUKE-4/2020 "Application-oriented education in ISO 9001:2015 requirements implementation" and "University Science Park TECHNICOM for Innovation Application Supported by Knowledge Technology", ITMS: 26220220182, supported by the Research & Development Operational Program funded by the ERDF.

**Institutional Review Board Statement:** Not applicable.

**Informed Consent Statement:** Not applicable.

**Data Availability Statement:** Not applicable.

**Acknowledgments:** Not applicable.

**Conflicts of Interest:** The authors declare no conflict of interest.

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
