# Peer review of "Improving Quality in the Process of Hot Rolling of Steel Sheets"

_applsci, doi:10.3390/app11125451_

Round 1

Reviewer 1 Report

Reviewer remarks to the article:

Improving Quality in the Process of Hot Rolling of Steel Sheets

This paper is well constructed. The results of investigations are valuable and very interesting from the point of view of the hot rolling of steel sheets process development. This is the correct planned and done scientific work. The authors applied a methodical apparatus adequate to the assumed goals. But the following comments should be addressed before considering of publication:

  • English should be checked.
  • There are some typographical errors.
  • No data available on the material used during the research.

In my opinion, Conclusions are somehow simplistic as it seems to be observational without revealing findings of generic academic value. What I mean that based on the results some generic and fundamental academic conclusions need to be drawn. In Conclusions, please try to emphasise the novelty, put some quantifications and comment on the limitations. The conclusions should also highlight the progress in understanding of knowledge presented in the work.

Author Response

I attach my comments to Reviewer1 in a Word file (REW_RESPONSE 1 applsci-1235479).

Reviewer 2 Report

The manuscript entitled ”Improving Quality in the Process of Hot Rolling of Steel Sheets” done by Stefan Markulik, Anna Nagyova, Renata Turisova, and Tomas Villinsky treats about improvements in process of steel sheets production. It seems that the authors propose changes in the technological process allowing to save 305 k€ per 18 weeks of production by excluding the most frequent errors. In my opinion, these findings appear to be important. However, I do not understand why they give up further research related to finding proper material for guide rails. In the manuscript, they claim it is, in their opinion, the main issue causing faults. It cannot be an economical one, as the savings have to bigger. Despite this, the manuscript needs some more accurate adjustments.

  • First, introducing the reference like “In [x], …” where the authors want to cite someone's findings, please call at least the name of the first author!
  • Between lines 98 – 111 the division into subparagraphs is not allowed. There is a clarification of the criteria allowing to distinguish different defects. The method of pairwise comparison is not cleared.
  • Line 148 “In this case, the number 1…” in which one? There weren’t shown any conditions allowing assigned value 1 or 0 to a proper case in Table 2! In my opinion, values not numbers are assigned.
  • Figure 3 and their explanation in line 195. In the Figure are not assigned any peaks to the Pd! I’m surprised by the results of the EDS investigation. Could you explain why the gold is presented in this defect? Peak’s intensities of Au and Fe are comparable!
  • Related to results presented in Figures 6 and 8, they are not understandable. What is the meaning of H0, H6, and so on?

Author Response

I attach my comments to Reviewer2 in a Word file (REW_RESPONSE 2 applsci-1235479).

Round 2

Reviewer 2 Report

Dear authors,

I do not agree with your explanations. In my opinion, the manuscript hasn’t been changed a lot.

Firstly,  you have not explained the condition when value 1 or 0 has to be assigned to the proper cause in Table 2!

Secondly, I have not found   an improved version of the Figure 3. Of course EDS spectrum ranges in wide energies. It reflects the specific radiation of the elements.   In Figure 3 there is assigned peak with a high intensity to the gold element, indeed. It means, in these defect the gold is present. And my question is: what is the origin of such defects where the Au and Pd are present as you claim in the text and in Figure 3?   It is just not clear for readers.

Please introduce your explanations, in present version I cannot find the manuscript is ready for publication.

Author Response

I attached my response in word document. Thank you mery much.

Round 3

Reviewer 2 Report

After changes introduced to the manuscript, I accept their explanations and agree to the publication of achievements.